# MGM-Omni: Scaling Omni LLMs to Personalized Long-Horizon Speech

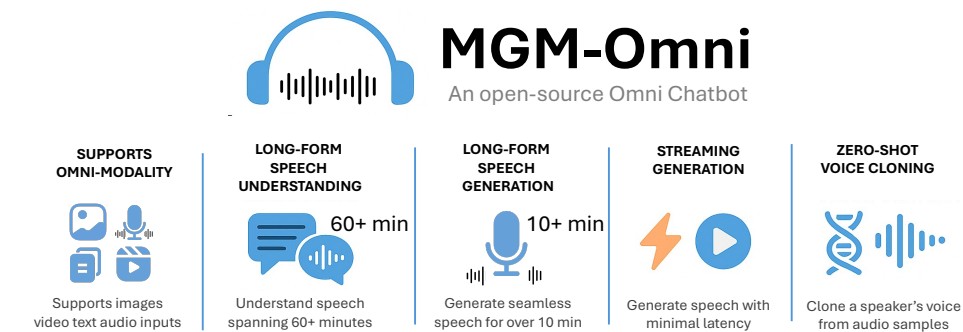

Figure 1: MGM-Omni is an advanced Omni LLM for omnimodal understanding, long-form understanding, long-form speech generation and zero-shot voice clone. It can comprehend audio inputs exceeding 60 minutes and produce consistent, high-quality speech outputs longer than 10 minutes.

## ABSTRACT

We present MGM-Omni, a unified Omni LLM for omni-modal understanding and expressive, long-horizon speech generation. Unlike cascaded pipelines that isolate speech synthesis, MGM-Omni adopts a "brain–mouth" design with a dual-track, token-based architecture that cleanly decouples multimodal reasoning from real-time speech generation. This design enables efficient cross-modal interaction and low-latency, streaming speech generation. For understanding, a unified training strategy coupled with a dual audio encoder design enables long-form audio perception across diverse acoustic conditions. For generation, a chunk-based parallel decoding scheme narrows the text–speech token-rate gap, accelerating inference and supporting streaming zero-shot voice cloning with stable timbre over extended durations. Compared to concurrent work, MGM-Omni achieves these capabilities with markedly data-efficient training. Extensive experiments demonstrate that MGM-Omni outperforms existing open source models in preserving timbre identity across extended sequences, producing natural and context-aware speech, and achieving superior long-form audio and omnimodal understanding. MGM-Omni establishes an efficient, end-to-end paradigm for omnimodal understanding and controllable, personalized long-horizon speech generation.

## 1 INTRODUCTION

The evolution of large language models (LLMs) from purely text-based systems (OpenAI, 2023a; Touvron et al., 2023) to multimodal frameworks has marked a significant paradigm shift in artificial intelligence. Vision language models (VLMs) such as LLaVA, GPT-4V, and Gemini (Liu et al., 2023b; OpenAI, 2023b; Team et al., 2023) have demonstrated remarkable capabilities in understanding and processing visual information, effectively bridging the gap between vision and language. Audio serves as a bridge between humans and AI. However, integration of audio, particularly understanding and generating long-form and expressive audio, remains a significant challenge in multimodal systems. Most existing approaches are vision-centric, treating audio as a secondary input modality and relying on separate, cascaded text-to-speech (TTS) systems for generation (Van

Den Oord et al., 2016; Anastassiou et al., 2024; Du et al., 2024). These methods exhibit critical shortcomings, including limited capability to process and understand extended audio sequences, high latency in audio synthesis, and degraded vocal timbre consistency over long durations.

The integration of audio in multimodal systems is hindered by the disparity between audio and text modalities. Audio token sequences are significantly more extensive and operate at a finer temporal resolution compared to their corresponding text token sequences (Van Den Oord et al., 2016; Shen et al.,

| Model | VU | AU | LAU | SG | LSG | VC |
|---|---|---|---|---|---|---|
| CosyVoice2 (Du et al., 2024) | | | | ✓ | | ✓ |
| Higgs-Audio-v2 (Boson AI, 2025) | | | | ✓ | ✓ | ✓ |
| Qwen2.5-VL (Bai et al., 2025) | ✓ | | | | | |
| Qwen2.5-Omni (Xu et al., 2025) | ✓ | ✓ | | ✓ | | |
| Lyra (Zhong et al., 2024) | ✓ | ✓ | ✓ | ✓ | | |
| MGM-Omni | ✓ | ✓ | ✓ | ✓ | ✓ | ✓ |

Table 1: **Function comparison.** VU, AU, LAU, SG, LSG, and VC denote visual understanding, audio understanding, long audio understanding, speech generation, long speech generation, and zero-shot voice cloning.

2018). This disparity creates three challenges. First, existing systems lack robust long-form audio understanding, struggling to maintain contextual coherence and semantic accuracy across extended audio inputs. Second, in generation, a one-to-many alignment problem complicates mapping semantic words or units to long acoustic sequences, leading to misaligned prosody and unnatural pacing in long-form speech. Third, the autoregressive generation process is prone to error accumulation, where minor inaccuracies cascade, degrading timbre consistency and audio quality. Despite recent progress (Huang et al., 2025; Xu et al., 2025; Team, 2025; Boson AI, 2025), these systems do not address the intertwined issues of long-form audio understanding, alignment, and generation.

To address these limitations, we introduce MGM-Omni, an Omni LLM that unifies vision, language, and audio in an end-to-end framework for seamless, low-latency multimodal understanding and generation. MGM-Omni adopts a dual-track architecture, separating multimodal reasoning (MLLM, the brain) from speech synthesis (SpeechLM, the mouth), enabling efficient cross-modal processing and real-time audio generation. For audio understanding, we employ a dual-encoder design that fuses acoustic and semantic features, with unified training enabling unified inference across short and long audio. For speech generation, we introduce Chunk-Based Parallel Decoding, which mitigates the token-rate gap between text and speech by segmenting text and predicting multiple speech tokens in parallel. This improves multimodal alignment, reduces long-sequence error accumulation and boosts inference speed by up to 3×. Trained on approximately 400k hours of audio, MGM-Omni supports zero-shot voice cloning from any personalized reference voice. Furthermore, we propose Long-TTS-Eval, a benchmark that systematically assesses long-form speech generation capability. Consequently, MGM-Omni delivers zero-shot voice cloning and expressive, personalized long-horizon speech, maintaining timbre consistency and robust text–speech alignment across extended contexts. Our main contributions are threefold:

- We propose MGM-Omni, an Omni LLM featuring a novel dual-track design that unifies omni-modal understanding and expressive speech generation, moving beyond cascaded systems.
- We introduce a Chunk-Based Parallel Decoding mechanism that mitigates the token-rate mismatch between text and speech, enabling efficient, high-fidelity, and context-aware long-form audio synthesis with customized voice.
- Through extensive experiments, we demonstrate that MGM-Omni significantly outperforms existing methods in long audio understanding, and achieves leading performance in zero-shot voice cloning and natural, context-aware long-form speech generation.

## 2 RELATED WORK

**Multi-modal Large Language Models.** The advent of large language models (LLMs) (OpenAI, 2023a; Touvron et al., 2023) has revolutionized natural language processing, paving the way for multimodal extensions that integrate diverse data modalities such as text, image, video and audio (Bai et al., 2025; Xu et al., 2025; Li et al., 2024b;c; Liu et al., 2023b; 2025a). Early multimodal models centered on vision–language alignment via contrastive learning. CLIP (Radford et al., 2021) demonstrated the efficacy of zero-shot image classification through joint embedding spaces. Building on this foundation, vision language models (VLMs) like Flamingo, LLaVA and MiniGPT-4 (Alayrac et al., 2022; Liu et al., 2023b; Zhu et al., 2023) adapted frozen visual encoders (e.g., CLIP-ViT) to

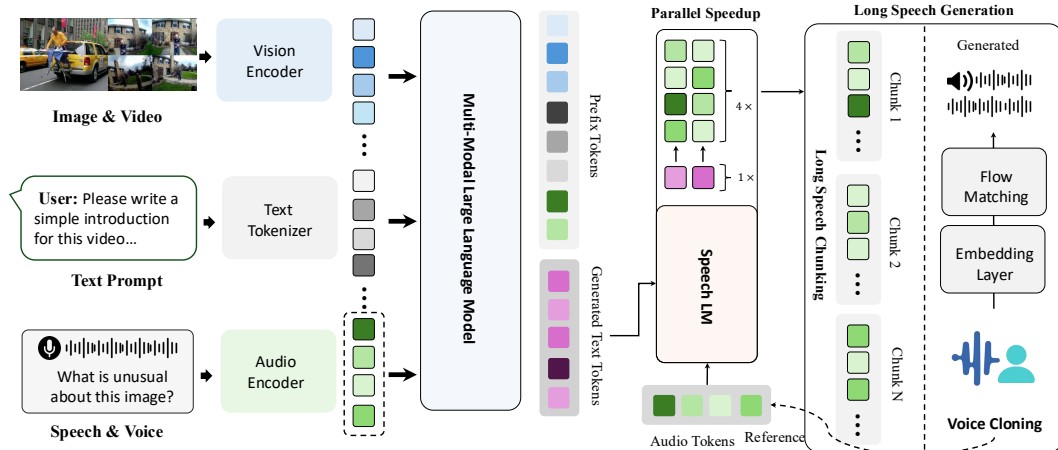

Figure 2: **The overview of MGM-Omni.** MGM-Omni decouples omni-modal understanding and speech generation into an MLLM and a SpeechLM. The MLLM processes text, images, video, and audio to produce text, while the SpeechLM generates speech from the MLLM's output in real time.

instruction-tuned LLMs to enable general-purpose multimodal understanding. Subsequent works such as Mini-Gemini (Li et al., 2024c), the LLaVA series (Liu et al., 2023a; Li et al., 2024a), and the Qwen-VL series (Wang et al., 2024; Bai et al., 2025) further advance VLMs with high-resolution image comprehension, video understanding and visual grounding. Despite this progress, most MLLMs remain vision-centric, with limited support for audio modalities. Recent efforts (Fu et al., 2024; Zhong et al., 2024; Xu et al., 2025) start to incorporate audio into MLLMs, but still struggle with understanding and generation of long-form audio, and cannot control the timbre of generated speech. MGM-Omni address these limitations with a dual-track, token-based architecture that natively fuses language and audio, enabling omni-modal understanding and expressive, controllable long-form audio generation.

**Speech Generation.** In recent years, driven by the emergence of large language models (LLMs) and large-scale speech-text pre-training, zero-shot text-to-speech generation (TTS) has advanced markedly (Anastassiou et al., 2024; Du et al., 2024; Boson AI, 2025). CosyVoice2 (Du et al., 2024) builds a TTS system with chunk-aware flow matching and LLMs, enabling streaming multilingual speech synthesis with zero-shot voice cloning. Qwen2.5-Omni (Xu et al., 2025) incorporates this design with a thinker–talker pipeline for end-to-end perception and generation across text, images, audio, and video. However, these systems still struggle with long-form speech generation. More recent efforts such as MOSS-TTSD (Team, 2025) and Higgs-Audio-v2 (Boson AI, 2025) support expressive bilingual dialogue generation with personalized voice, yet challenges remain in maintaining timbre consistency over long sequences, ensuring real-time cross-modal fidelity, and achieving low latency. MGM-Omni addresses this issue via a chunk-based parallel decoding approach, enabling expressive long-form speech generation with consistent timbre and low latency.

## 3 MGM-OMNI

MGM-Omni is capable of processing text, images, video and speech, and can generate both textual and spoken outputs. To support high-quality, long-form speech synthesis without compromising the efficiency and effectiveness of omnimodal understanding and text generation, MGM-Omni decouples multimodal understanding and speech generation into two components: MLLM, serving as the "brain" for multimodal understanding and text generation, and SpeechLM, serving as the "mouth" for real-time speech generation. For input in different modalities, we employ modality-specific encoders to extract features, which are subsequently passed to the MLLM. The MLLM generates text tokens and passes them to SpeechLM, which produces speech tokens in real-time via a Chunk-Based Parallel Decoding strategy. These speech tokens are further converted into Mel-spectrograms through a flow-matching model (Lipman et al., 2022), and the final audio is synthesized using a vocoder. The overall framework is illustrated in Figure 2.

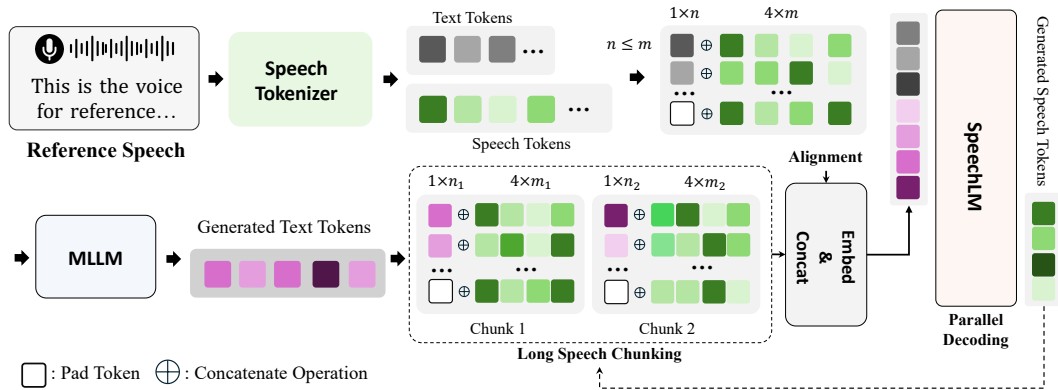

Figure 3: **The overview of SpeechLM in MGM-Omni.** Conditioned on MLLM-generated text and the reference audio clip, SpeechLM generate speech with Chunk-based Parallel Decoding.

## 3.1 OMNI UNDERSTANDING

MGM-Omni is built upon Qwen2.5-VL (Bai et al., 2025), a state-of-the-art open-source Vision-Language Model (VLM) that supports image and video understanding with a native-resolution ViT (Dehghani et al., 2023). Based on Qwen2.5-VL, MGM-Omni attempts to extend towards Omni-Understanding, especially by incorporating audio understanding capabilities.

**Dual Audio Encoder.** MGM-Omni adopts a dual audio encoder design to capture both acoustic and semantic audio features. The primary encoder, Qwen2-Audio (Chu et al., 2024), is an audio encoder continually trained on Whisper-large-v3 (Radford et al., 2022) for enhanced general sound perception. To strengthen semantic understanding, especially for Chinese speech, we incorporate Belle-Whisper-large-v3 (BELLEGroup, 2023), another Whisper-based encoder specialized in Chinese speech recognition. This dual encoder setup yields two complementary representations: the main audio feature $X_{\text{main}}$ and the auxiliary audio feature $X_{\text{aux}}$.

**Information Mining.** To effectively integrate these complementary features, we design an audio information mining approach inspired by Mini-Gemini (Li et al., 2024c). Specifically, $X_{\text{main}}$ serves as the query $Q \in \mathbb{R}^{N \times C}$, while $X_{\text{aux}}$ provides the key-value pair: $K \in \mathbb{R}^{N \times C}$ and $V \in \mathbb{R}^{N \times C}$, allowing the model to retrieve semantically relevant cues from $X_{\text{aux}}$ under the guidance of $X_{\text{main}}$. Formally, information mining can be defined as:

$$T_A = \text{MLP}(Q + \text{Softmax}(\phi(Q) \times \phi(K)^\top) \times \phi(V)), \tag{1}$$

where $\phi$ denotes a projection layer and MLP represents a multi-layer perceptron. This approach enhances the audio representation by making it both acoustically and semantically aware, yielding enhanced audio tokens $T_A$ for subsequent LLM processing.

**Training Strategy.** Following Lyra (Zhong et al., 2024), we build a two-stage training pipeline to integrate audio understanding capabilities. In the first stage, we conduct audio-to-text pre-training to align the audio encoder to LLM. In the second stage, we perform unified omni-modal training. The first stage primarily uses audio transcription data, while the second stage comprises audio transcription, audio QA, audio-instruct VQA, and text instruction tuning data. This training strategy enables omni-cognition and robust cross-modal reasoning.

**Omni Length Understanding.** MGM-Omni aims to support both long and short sequence input. However, training with sequences of diverse lengths is inefficient: large batch sizes cause long sequence samples to run out of memory, while small sizes waste memory on short sequence samples. To address this issue, we propose a unified training pipeline. First, we group audio of similar lengths into the same batch. Second, we dynamically adjust the batch size, smaller for long-context inputs and larger for short-context inputs. This strategy significantly improves training efficiency.

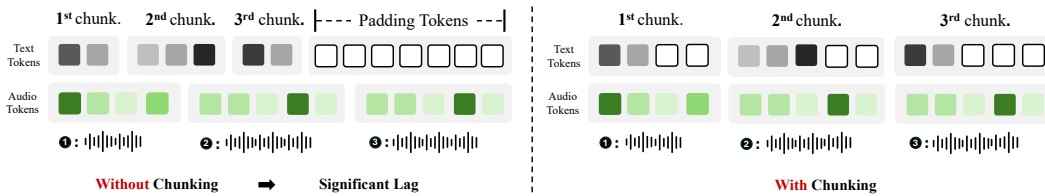

Figure 4: **Decoding comparison.** Chunk-based decoding narrows the gap between text and corresponding speech, enabling long-form speech generation.

## 3.2 OMNI GENERATION

MGM-Omni can generate both long-form text and speech. The textual output is autoregressively produced by the Omni-MLLM. The generated text, together with the personalize reference audio, is subsequently served as the conditioning for SpeechLM to synthesize speech via a Chunk-based Parallel Decoding method. The overall speech generation pipeline is depicted in Figure 3.

**Speech Generation.** SpeechLM takes text tokens from Omni-MLLM as input and generates speech tokens in an autoregressive manner. It is initialized from the Qwen3 (Yang et al., 2025a) language model, with an additional TTS-Adapter appended to its output. TTS-Adapter consists of six randomly initialized Qwen3 blocks, designed to transform text representations into speech representations. The speech tokens produced by SpeechLM are then converted into Mel-spectrograms through a Flow-Matching model, and finally synthesized into audio via HiFi-GAN (Kong et al., 2020) vocoder. We used the flow-matching model from CosyVoice2 (Du et al., 2024), which supports chunk-aware streaming decoding.

**Speech Tokenizer.** We employ the CosyVoice2 finite scalar quantization (FSQ) speech tokenizer to obtain discrete speech representations for speech generation. The tokenizer operates at a rate of 25 Hz, meaning that 25 tokens represent one second of audio. In comparison, humans typically express only two or three words per second. This discrepancy highlights that for a given utterance, the number of speech tokens is substantially larger than the number of text tokens. This leads to the following two issues:

- As the length of the speech increases, the gap between text and speech tokens widens, weakening their correlation and degrading the quality of long-form generation.
- The much higher number of speech tokens compared to text tokens slows inference and harms streaming efficiency.

To address these two challenges, we propose a Chunk-Based Parallel Decoding for efficient long-form speech generation.

**Chunk-based Decoding.** To improve text–speech alignment in long-form speech generation, we introduce Chunk-based Decoding for speech token generation. As shown in Figure 4, the input text is divided into smaller chunks that are sequentially processed by SpeechLM, with each chunk producing a corresponding speech segment. During decoding, we adopt a token delay strategy: speech token generation within a chunk is initiated only after the first four text tokens, which are replaced by padding tokens in the speech sequence. This design ensures that every speech token is aligned with its corresponding text token while avoiding early mis-synchronization. By reducing the alignment distance between modalities, Chunk-based Decoding enhances cross-modal correspondence and improves the robustness of long-form speech synthesis. In contrast to naive segmentation methods, our approach preserves both the previously generated text and speech as context, thereby maintaining global fluency and coherence in the final output. Notably, Chunk-based Decoding is highly compatible with our dual-track "brain–mouth" design, preserving omnimodal understanding and text generation speed while improving speech synthesis quality.

**Parallel Decoding.** To improve efficiency, we introduce a parallel decoding strategy for efficient speech token generation. Specifically, we extend the vocabulary so that SpeechLM can decode both modalities in a single step. Let $V_{\text{text}}$ denote the text vocabulary, $V_{\text{speech}}$ denote the speech tokenizer

vocabulary, and $k$ denote the parallel size. The extended vocabulary size is thus defined as:

$$|V| = |V_{\text{text}}| + k|V_{\text{speech}}|. \tag{2}$$

For speech tokenization, the input for each decoding step $t$ consists of one text token $x_t$ and $k$ speech tokens $\{s_t^1, s_t^2, \ldots, s_t^k\}$. We use $f(\cdot)$ to denote the embedding function, and the hidden features $h_t^{in}$ for LLM input can be averaged as:

$$h_t^{in} = \frac{1}{k+1} \left( f(x_t) + \sum_{i=1}^{k} f(s_t^i) \right). \tag{3}$$

For speech detokenization, we employ a TTS-Adapter to project the LLM output hidden state $h_t^{out}$ into the speech representation space, after which the lm_head predicts the next set of speech tokens:

$$\{\hat{s}_{t+1}^1, \ldots, \hat{s}_{t+1}^k\} = \text{lm\_head}\big(\text{TTS-Adapter}(h_t^{out})\big). \tag{4}$$

While parallel decoding is commonly used with RVQ speech tokenizers (Xie & Wu, 2024a; Team, 2025), it is rarely applied to FSQ speech tokenizers. We found that using parallel decoding with FSQ speech tokenizers not only maintains speech synthesis performance but also significantly improves efficiency. Additionally, it further shortens the distance between text and speech tokens, enhancing their correlation.

## 3.3 OMNI VOICE

MGM-Omni is capable of generating long-form speech in any personalized voice. To enable this capability, we carefully designed both the data pipeline and the training strategy.

**Training Data.** To enable zero-shot voice cloning, we collected a large-scale dataset, including around 300k hours of raw speech data and approximately 100k hours of TTS-synthesized speech in Chinese and English. The raw speech portion of our corpus incorporates diverse open-source datasets, including Emilia Dataset (He et al., 2024), Libri-heavy (Kang et al., 2024), Common Voice (Ardila et al., 2019), and Aishell series (Bu et al., 2017; Du et al., 2018; Shi et al., 2020). We constructed a dataset for TTS synthesis by sampling Chinese conversations from Belle-10M (BEL-LEGroup, 2023) and English conversations from Lamini-Instruct (Wu et al., 2023). We uniformly sampled 900k Chinese and 700k English conversations based on length. As these datasets are somewhat outdated, we enhanced the text quality by regenerating all responses using Qwen2.5-72B (Yang et al., 2025b). Subsequently, we synthesized audio from these refined conversations using megatts3 (Jiang et al., 2025). For each sample, we randomly select a reference voice from the provided set of pre-processed reference audio.

**Pre-training.** The SpeechLM consists of a pre-trained Qwen3 (Yang et al., 2025a) LLM paired with a randomly initialized TTS-Adapter. The model is trained to generate speech from given text and reference audio through a next speech token prediction objective. The goal of the pre-training stage is to align the speech and text modalities. At this stage, the parameters of the pre-trained Qwen3 LLM remain frozen, while only the TTS-Adapter is updated. Both raw and synthesized speech data are leveraged in pre-training to ensure robustness across diverse speaker timbres.

**Post-training.** The post-training phase aims to enhance SpeechLM's capacity for fluent and accurate speech generation. During this phase, the parameters of both the LLM and the TTS-Adapter are jointly optimized with different learning rates. The TTS-Adapter is trained at a rate five times higher than that of the LLM. The training corpus is primarily composed of high-fidelity TTS-synthesised speech, supplemented with a smaller portion of raw speech data.

## 4 EXPERIMENTS

### 4.1 MAIN PROPERTIES

In this section, we present a comprehensive evaluation covering audio understanding, omni-modality understanding, and speech generation, to demonstrate the main properties of MGM-Omni, with particular emphasis on its capacity for long audio understanding and long audio generation and zero-shot voice cloning.

| Model | LibriSpeech Test | | CommonVoice | | AISHELL |
| --- | --- | --- | --- | --- | --- |
| | clean WER ↓ | other WER ↓ | EN WER ↓ | ZH CER ↓ | CER ↓ |
| *Audio LLMs* | | | | | |
| Whisper-large-v3 (Radford et al., 2022) | 1.8 | 3.6 | 9.3 | 12.8 | |
| Qwen2-Audio (Chu et al., 2024) | **1.3** | **3.4** | **8.6** | **5.2** | |
| *Omni LLMs* | | | | | |
| Mini-Omni2 (Xie & Wu, 2024b) | 4.7 | 9.4 | | | |
| Lyra (Zhong et al., 2024) | 2.0 | 4.0 | | | |
| VITA-1.5 (Fu et al., 2024) | 3.4 | 7.5 | | | 2.2 |
| Ola (Liu et al., 2025b) | 1.9 | 4.3 | | | |
| Qwen2.5-Omni (Xu et al., 2025) | 1.6 | 3.5 | **7.6** | 5.2 | |
| MGM-Omni-7B | 1.7 | 3.6 | 8.8 | 4.5 | 1.9 |
| MGM-Omni-32B | **1.5** | **3.2** | 8.0 | **4.0** | **1.8** |

Table 2: **Omni-comparison on ASR benchmarks.** We use Common-Voice, LibriSpeech and AISHELL to evaluate the ASR capability on Chinese and English.

| Model | Speech ↑ | Sound ↑ | Music ↑ | Mix ↑ | Average ↑ |
| --- | --- | --- | --- | --- | --- |
| *Audio LLMs* | | | | | |
| SpeechGPT (Zhang et al., 2023) | 1.6 | 1.0 | 1.0 | 4.1 | 1.9 |
| SALMONN (Tang et al., 2023) | 6.2 | 6.3 | 6.0 | 6.1 | 6.1 |
| Qwen2-Audio (Chu et al., 2024) | **7.2** | **7.0** | **6.8** | **6.8** | **6.9** |
| *Omni LLMs* | | | | | |
| LLaMA-Omni (Fang et al., 2024) | 5.2 | 5.3 | 4.3 | 4.0 | 4.7 |
| Mini-Omni2 (Xie & Wu, 2024a) | 3.6 | 3.5 | 2.6 | 3.1 | 3.2 |
| IXC2.5-OmniLive (Zhang et al., 2024) | 1.6 | 1.8 | 1.7 | 1.6 | 1.7 |
| VITA-1.5 (Fu et al., 2024) | 4.8 | 5.5 | 4.9 | 2.9 | 4.5 |
| Qwen2.5-Omni (Xu et al., 2025) | 6.8 | 5.7 | 4.8 | 5.4 | 5.7 |
| Ola (Liu et al., 2025b) | **7.3** | 6.4 | 5.9 | 6.0 | 6.4 |
| MGM-Omni-7B | **7.3** | **6.5** | **6.3** | 6.1 | **6.5** |
| MGM-Omni-32B | 7.1 | **6.5** | 6.2 | **6.2** | **6.5** |

Table 3: **Omni-comparison on Audio QA benchmarks.** We use AIR-Bench for audio QA evaluation. The scores are evaluated by gpt-4-0125-preview.

### 4.1.1 AUDIO UNDERSTANDING

**Short Audio Understanding.** We compare the audio understanding ability (audio → text) of MGM-Omni against leading Audio and Omni LLMs on two primary tasks: automatic speech recognition (ASR) and general audio QA. First, we evaluate the ASR ability on LibriSpeech (Panayotov et al., 2015), CommonVoice (Ardila et al., 2019) and AISHELL (Bu et al., 2017). As shown in Table 2, MGM-Omni delivers competitive or superior performance for both English and Chinese ASR. In particular, with dual audio encoder, MGM-Omni achieves 4.0 CER on CommonVoice (ZH) and 1.8 CER on AISHELL, surpassing leading audio and Omni LLMs. For general audio understanding, we evaluate audio QA on AIR-Bench (Yang et al., 2024), a comprehensive benchmark covering speech, sound, and music inputs. As summarized in Table 3, MGM-Omni outperforms all open source Omni LLMs, including Qwen2.5-Omni (Xu et al., 2025).

**Long Audio Understanding.** Unlike many open-source Audio and Omni LLMs, MGM-Omni is capable of processing audio inputs exceeding one hour in length. To evaluate its ability on long-form audio understanding, we conducted a needle-in-the-haystack test. As illustrated in Figure 5, MGM-Omni successfully handles audio inputs of up to 4,500 seconds, significantly outperforming Qwen2.5-Omni (Xu et al., 2025). The success rate is averaged over five diverse long-form audio. Moreover, we provide quantitative comparison in Figure 7 in the appendix.

### 4.1.2 OMNI-MODALITY UNDERSTANDING

MGM-Omni processes text, image, video, and audio inputs. Following Lyra (Zhong et al., 2024), we further evaluate its omni-modal understanding (multimodality → text) by comparing MGM-Omni against other omni-modal LLMs on several speech-instructed VQA benchmarks. As shown in Table 4, MGM-Omni shows a strong ability to follow speech instructions.

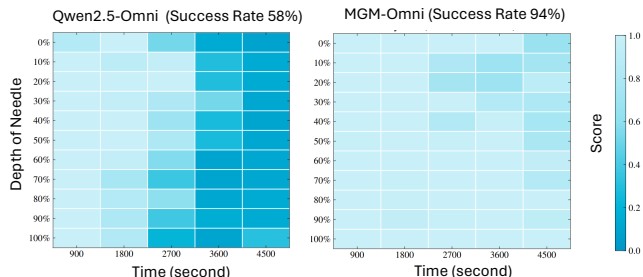

Figure 5: **Omni-comparison for Long-form Audio.** We adopt a needle-in-the-haystack evaluation and report the average success rate across five materials.

| Model | TextVQA-Speech ↑ | DocVQA-Speech ↑ | ChartVQA-Speech ↑ | AI2D-Speech ↑ |
|---|---|---|---|---|
| Intern-Omni-9B (OpenGVLab, 2024) | 69.1 | 80.0 | 56.1 | 54.0 |
| Lyra-9B (Zhong et al., 2024) | 80.0 | 85.5 | 61.0 | 63.1 |
| MGM-Omni-7B | **81.7** | 87.4 | 69.3 | 70.4 |
| MGM-Omni-32B | 78.2 | **88.4** | **72.1** | **71.3** |

Table 4: **Omni-comparison on vision-speech benchmarks.** We convert the textual questions in multiple VQA benchmarks into synthesized speech to evaluate the multimodal understanding ability.

### 4.1.3 SPEECH GENERATION

MGM-Omni supports long-form synthesis (exceeding 10 minutes) with customizable voices. Here, we assess the speech generation capabilities (text → speech) in both short- and long-form setting.

**Short Speech Generation.** We evaluated MGM-Omni against state-of-the-art zero-shot TTS systems and Omni LLMs to assess the speech generation capabilities. As shown in Table 5a, MGM achieves lower error rates and higher speaker similarity than open-source TTS models and Omni LLMs on seed-tts-eval (Anastassiou et al., 2024), demonstrating strong text-to-speech performance and robust zero-shot voice cloning.

**Long Speech Generation.** Unlike many open-source Omni LLMs and TTS systems, MGM-Omni can generate over 10 minutes of speech in any personalized voice. Quantitative examples are shown in Figure 8 in the appendix. For benchmark evaluation, most existing benchmarks only evaluate short clips, typically ranging from a few seconds to a few dozen seconds, leaving a gap in assessing long-form performance. Moreover, existing TTS benchmarks focus on normal text generation and do not cover more complex text, such as formulas, URLs, or classical Chinese poetry. To address this, we introduce **Long-TTS-Eval**, a benchmark specifically designed to evaluate long-form text-to-speech generation systematically. We leave more detailed information about the benchmark to Section A.2 in the appendix.

We compare MGM-Omni against two categories of open-source TTS systems: (1) Native long TTS models, represented by MOSS-TTSD-v0.5 (Team, 2025) and Higgs-Audio-v2 (Boson AI, 2025). (2) Non-native models that extend via chunking, represented by CosyVoice2 (Du et al., 2024). We report WER for English TTS, CER for Chinese TTS, and RTF for inference efficiency. As shown in Table 5b, MGM-Omni achieves lower error rates across most speech generation scenarios, along with the lowest RTF. It is worth noting that, MGM-Omni's two-stage training relies on less than 400k hours of audio, substantially fewer than the 1M or even 10M hours used in concurrent works. This result demonstrates the efficiency, effectiveness, robustness and data efficiency of our model.

### 4.2 ABLATION STUDY

**Audio Encoder.** We ablate different audio encoder designs and evaluate on CommonVoice ASR (Ardila et al., 2019). As shown in Table 6a, incorporating both the Qwen2-Audio encoder (Chu et al., 2024) and the Belle-Whisper-large-v3 encoder (BELLEGroup, 2023) with information mining yields the best performance in audio understanding. Note that, compared with the final model, we do not use the long audio QA data here.

| Model | Size | EN WER ↓ | EN SIM ↑ | ZH CER ↓ | ZH SIM ↑ |
|---|---|---|---|---|---|
| CosyVoice2 (Du et al., 2024) | 0.5B | 2.57 | 0.652 | 1.45 | 0.748 |
| Qwen2.5-Omni-3B (Xu et al., 2025) | 0.5B | 2.51 | 0.635 | 1.58 | 0.744 |
| Qwen2.5-Omni-7B (Xu et al., 2025) | 2B | 2.33 | 0.641 | 1.42 | 0.754 |
| Higgs-Audio-v2 (Boson AI, 2025) | 6B | 2.44 | 0.677 | 1.66 | 0.743 |
| MGM-Omni-TTS-0.6B | 0.6B | 2.48 | 0.670 | 1.42 | 0.750 |
| MGM-Omni-TTS-2B | 2B | 2.28 | 0.684 | 1.28 | 0.755 |
| MGM-Omni-TTS-4B | 4B | **2.22** | **0.686** | **1.18** | **0.758** |

(a) Zero-shot short TTS comparison of error rate and speaker similarity in Seed-TTS-Eval. For Qwen2.5-Omni, size indicates the talker module size.

| Model | Size | RTF ↓ | EN WER ↓ | ZH CER ↓ | EN-hard WER ↓ | ZH-hard CER ↓ |
|---|---|---|---|---|---|---|
| CosyVoice2 (chunk) (Du et al., 2024) | 0.5B | 0.34 | 14.80 | **5.27** | 42.48 | 32.76 |
| MOSS-TTSD-v0.5 (Team, 2025) | 2B | 0.23 | 8.69 | 6.82 | 62.61 | 62.97 |
| Higgs-Audio-v2 (Boson AI, 2025) | 6B | 0.33 | 27.09 | 31.39 | 98.61 | 98.85 |
| MGM-Omni-TTS-2B | 2B | **0.19** | **4.98** | 5.58 | **26.26** | **23.58** |

(b) Long-form TTS comparison of error rate and inference speed in our Long-TTS-Eval.

Table 5: **Omni-comparison TTS benchmarks.** We evaluate short-form and long-form TTS using Seed-TTS-Eval (top) and Long-TTS-Eval (bottom).

| Audio Encoder | EN WER ↓ | ZH CER ↓ |
|---|---|---|
| Qwen2-Audio | 13.0 | 3.9 |
| Belle-Whisper | 21.7 | 5.0 |
| Info Mining | **9.1** | **3.5** |

| Parallel | RTF | EN WER ↓ | ZH CER ↓ |
|---|---|---|---|
| 1 | 0.57 | **1.86** | **1.15** |
| 2 | 0.32 | 2.02 | 1.23 |
| 4 | 0.19 | 2.28 | 1.28 |

| Chunking | EN WER ↓ | ZH CER ↓ |
|---|---|---|
|  | 31.84 | 8.97 |
| ✓ | **4.98** | **5.64** |

| (a) Audio Encoder | (b) Parallel Decoding | (c) Chunk-Based Decoding |
|---|---|---|

Table 6: **Ablation study.** We conduct ablation studies on the audio encoder, parallel decoding, and chunk-based decoding.

**Chunk-based Decoding.** We evaluate long-form speech generation on our Long-TTS-Eval to assess the impact of chunk-based decoding. As shown in Table 6c, removing chunk-based decoding leads to a substantially higher error rate, exceeding that of concurrent works. Given that concurrent methods typically use millions to tens of millions of hours of audio, we attribute MGM-Omni's data efficiency primarily to its use of chunk-based decoding.

**Parallel Decoding.** We ablate the impact of parallel decoding by comparing both TTS performance and inference speed. TTS performance is measured on Seed-TTS-Eval (Anastassiou et al., 2024), while inference speed is assessed using 16 Chinese and 16 English samples drawn from Long-TTS-Eval. We report the real-time factor (RTF) on a single H800 GPU to compare the inference speed. As shown in Table 6b, increasing the parallel size slightly raises the audio error rate but substantially accelerates inference by 3×. To balance quality and speed, we set the parallel size to 4. We anticipate that incorporating more advanced Multi-Token Prediction (MTP) techniques (Liu et al., 2024) will further improve audio quality at larger parallel sizes.

## 5 CONCLUSION

We present MGM-Omni, a unified Omni LLM that supports long-form omnimodal understanding and robust long-duration speech generation with personalized voices. Its dual-track architecture separates multimodal reasoning (MLLM) from real-time speech synthesis (SpeechLM), enabling efficient cross-modal interaction within an end-to-end framework. For understanding, it employs a dual audio encoder that fuses acoustic and semantic cues, yielding robust long-form audio perception. For generation, we introduce Chunk-Based Parallel Decoding to bridge the token-rate gap between text and speech, enabling efficient, low-latency synthesis, while conditioning SpeechLM on reference audio to support zero-shot voice cloning with consistent timbre. Experiments show that MGM-Omni surpasses leading open source Omni LLMs in timbre consistency, context-aware speech, long audio comprehension, and omni-modal reasoning.

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

# A APPENDIX

## A.1 DATA FORMAT

The data format for MLLM and SpeechLM with the same instruction is illustrated in Figure 6. SpeechLM use chunk-based decoding to generate long-form speech.

---

**The Data Format of MLLM in MGM-Omni**

**<|im_start|>**user
Introduce the capital of France.
**<|im_end|>**

**<|im_start|>**assistant
The capital of France is Paris. ... is one of the most populous cities in Europe. Paris is known for its rich history, ... forits fashion, cuisine, and nightlife. Overall,Paris is a vibrant and dynamic city ... each year from around the world.
**<|im_end|>**

---

**The Data Format of SpeechLM in MGM-Omni**

**<|im_start|>**user
Respond with the tone of the reference audio clip.**<|audio_start|>** *[Transcript of the refer-ence audio]* **<|audio_end|>**
Introduce the capital of France.
**<|im_end|>**

**<|im_start|>**assistant
**<|audio_start|>**
The capital of France is Paris. ... is one of the most populous cities in
Europe.**<|end_of_text|>** ... **<|end_of_text|><|audio_sep|>**                                    **Chunk 1**

Paris is known for its rich history, ... forits fashion, cuisine, and nightlife. **<|end_of_text|>** ...
**<|end_of_text|>** **<|audio_sep|>**
                                                                                  **Chunk 2**

Overall,Paris is a vibrant and dynamic city ... each year from around the world.
**<|end_of_text|>** ...**<|end_of_text|>**
**<|audio_end|>**                                                                 **Chunk 3**
**<|im_end|>**

---

Figure 6: The data format of MLLM (top) and SpeechLM (button) in MGM-Omni.

## A.2 LONG-TTS-EVAL BENCHMARK

In this section, we provide a detailed introduction to the data composition and evaluation protocol of the Long-TTS-Eval benchmark we constructed.

### A.2.1 DATA COMPOSITION

Long-TTS-Eval focuses on assessing TTS systems' capabilities in long-form speech generation and complex case handling.

For long TTS evaluation, we collected six types of text: literature, news, knowledge, speeches, reviews, and academic papers, comprising 341 Chinese samples and 353 English samples. The data were sourced from news outlets, Wikipedia, YouTube video transcripts, and arXiv papers. We use the Qwen3 tokenizer (Yang et al., 2025a) to calculate the token length. As illustrated in Tabel 7, the maximum length is 1899 tokens in Chinese and 3277 tokens in English, and the average length

| Category | Samples (ZH) | Avg Length (ZH) | Max Length (ZH) | Samples (EN) | Avg Length (EN) | Max Length (EN) |
|---|---|---|---|---|---|---|
| Literature | 41 | 998.8 | 1644 | 56 | 985.5 | 1344 |
| News | 60 | 585.4 | 1159 | 60 | 915.4 | 1781 |
| Knowledge | 60 | 764.0 | 1279 | 59 | 1130.7 | 3245 |
| Talk | 60 | 619.8 | 1885 | 59 | 952.4 | 2745 |
| Comment | 60 | 513.8 | 1537 | 59 | 844.6 | 2096 |
| Paper | 60 | 753.5 | 1899 | 60 | 1281.2 | 3277 |
| Total | 341 | 689.6 | 1899 | 353 | 1019.0 | 3277 |

Table 7: The composition and average length of our Long-TTS-Eval benchmark.

| Category | Samples (ZH) | Avg Length (ZH) | Max Length (ZH) | Samples (EN) | Avg Length (EN) | Max Length (EN) |
|---|---|---|---|---|---|---|
| URLs | 57 | 96.8 | 180 | 45 | 102.5 | 166 |
| Emails | 45 | 63.6 | 97 | 44 | 85.5 | 136 |
| Phone | 30 | 92.0 | 160 | 30 | 117.2 | 199 |
| Number | 33 | 83.7 | 159 | 30 | 94.0 | 136 |
| Math | 100 | 606.8 | 955 | 100 | 605.8 | 1009 |
| Total | 265 | 281.4 | 955 | 260 | 293.9 | 1009 |

Table 8: The composition and average length of the hard set in our Long-TTS-Eval benchmark.

is 689.57 tokens in Chinese and 1019.0 tokens in English. As a single-point timing estimate, 1899 Chinese tokens correspond to about 10 minutes of speech (assuming 200 characters per minute and 1 token per character), and 3277 English tokens correspond to about 12 minutes (assuming 215 words per minute and 1.3–1.5 tokens per word).

For complex case handling, we collected five types of text: web URLs, emails, math formulas, phone numbers, and large numbers, comprising 265 Chinese samples and 260 English samples. The detailed information is illustrated in Tabel 8. Mathematical formulas were sourced from the reasoning process and solution from S1 Long-CoT Instruct dataset Muennighoff et al. (2025), while the other categories were generated by Gemini 2.5 Pro (Comanici et al., 2025).

### A.2.2 EVALUATION PIPELINE

We follow Seed-TTS-Eval (Anastassiou et al., 2024) to build our evaluation pipeline. We use Whisper-large-v3 (Radford et al., 2022) and Paraformer-zh (Gao et al., 2023) as the automatic speech recognition (ASR) engines for English and Chinese, respectively. Since both models accept only short audio, we segment each generated waveform into 28-second chunks, transcribe each chunk independently, and then concatenate the transcripts to obtain the final transcription. We then compute word error rate (WER) for English and character error rate (CER) for Chinese.

### A.2.3 EVALUATION WITH NORMALIZED TEXT

Conventional TTS benchmarks often transcribe generated speech with an ASR model and then compare the transcript to the ground-truth text to calculate the error rate. This approach has a key flaw: for expressions with multiple valid readings, ASR outputs can legitimately differ from the written form. For example, "5%" spoken by TTS may be transcribed as "five percent." It differs from the ground truth, but it is still correct.

To address this issue, for each sample with ground-turth $G$, we prompt GPT-5 (OpenAI, 2025) to generate a normalized ground-turth $N$ that reflects a natural spoken version. We then synthesize speech, obtain the ASR transcript $T$, and compute two word error rates, between $T$ and $G$, and between $T$ and $N$. The final per-sample error is the smaller of the two:

$$WER_{sample} = min(WER(T, G), WER(T, N)) \tag{5}$$

This method lowers the risk of falsely flagging correct TTS, thereby enhancing the reliability of the reported error rates.

### A.3 Quantitative Results

### A.3.1 Long Audio Understanding

To verify MGM-Omni's effectiveness in long audio understanding, we conducted a more in-depth evaluation. We illustrate the quantitative result in Figure 7. For long audio summarization, MGM-Omni provides more complete and detailed responses compared with Qwen2.5-Omni (Xu et al., 2025). For fine-grained understanding, MGM-Omni accurately extracts information from long audio inputs, while Qwen2.5-Omni refuses to respond.

### A.3.2 Long Speech Generation

We compare MGM-Omni with concurrent long TTS systems, MOSS-TTSD-v0.5 (Team, 2025) and Higgs-Audio-v2 (Boson AI, 2025) to evaluate the long-form speech generation capability. Specifically, we evaluate two challenging pieces: the renowned Chinese long prose poem "Preface to the Pavilion of Prince Teng" (Tengwang Ge Xu) and Tagore's famous poem "Stray Birds" excerpt "Life is as ephemeral as summer flowers" featuring mixed Chinese–English code switching. As depicted in Figure 8, MGM-Omni produces accurate speech with appropriate pausing, while competing methods exhibit pronounced errors in the latter portions of the audio, including audible noise.

### A.4 Use of LLMs

In this paper, we use large language models (LLMs) to correct grammatical errors and polish our paper writing.

**Long-Speech Ability:** compare with the Qwen2.5-Omni

**20 mins** ABC News, From YouTube      **TASK: Summarize the News**

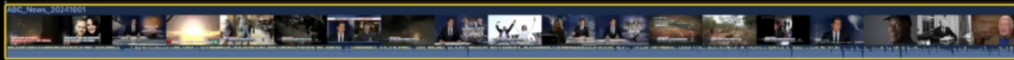

**MGM-Omni:**      **234 words with more details**

The audio is a news segment covering several major events and stories. It begins with a report *on Iran's attack on Israel*, describing it as the largest missile attack ever, involving over 200 ballistic missiles targeting Israeli airbases and intelligence headquarters. Israel's defenses and the US military intercepted many of these missiles, but Israel vowed to strike back. The segment also mentions a separate deadly shooting in Tel Aviv, where terrorists killed at least seven people. The situation raises concerns about a potential wider regional war, with Israel launching operations against Hezbollah infrastructure in Lebanon. The US is actively supporting Israel's defense efforts. The segment transitions to coverage of *the vice-presidential debate* between Governor Tim Walz and Senator J.D. Vance, which is scheduled for later that night. The debate is expected to be significant due to its impact on the election, particularly in close battleground states. The segment also discusses *the ongoing dock workers strike affecting crucial ports along the East Coast and Gulf Coast, causing supply chain disruptions and potential economic impacts*. Additionally, the segment covers the aftermath of *Hurricane Ida, highlighting the rising death* toll and ongoing search efforts for missing individuals. It also touches on a legal controversy involving Texas attorney Sean Diddy Combs, who faces accusations of sexual assault. The segment concludes with tributes to former *President Jimmy Carter on his 100th birthday, noting his humanitarian work and legacy.*

**Qwen2.5-Omni:**      **153 words**

The audio is about a big news story. It starts with a vice-presidential debate between Tim Walls and JD Vance. Then it talks about Iran attacking Israel with over 200 ballistic missiles. Israel is responding and vowing to strike back. There are scenes of missiles raining down on military bases and cities. One person was killed in the West Bank and several were injured. The US is closely monitoring the situation. *Israel is planning a response to Iran's attack*. There's also a separate shooting outside a Tel Aviv rail station. Israel launched a ground operation in southern Lebanon targeting Hezbollah infrastructure. Iran claims the attack is retaliation for Israel's actions. President Biden says the US supports Israel. Israel's Prime Minister Netanyahu said Iran made a big mistake. Israel is getting pressure to hit targets like Iranian oil infrastructure. *So, what do you think about all this? Do you have any other questions?*

**Long-Speech Ability:** compare with the Qwen2.5-Omni

**Question:** How many people died in the hurricane?      **TASK: extract detailed answers from long speech**

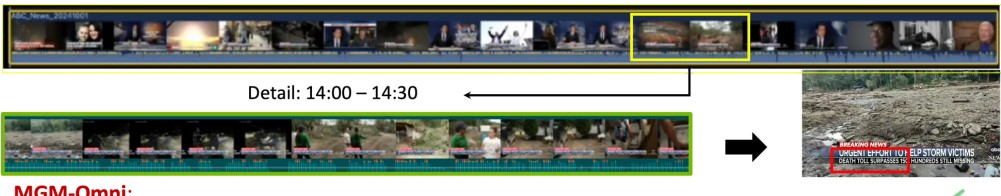

Detail: 14:00 – 14:30

**MGM-Omni:**

……
The death toll from *Hurricane Ida has risen to more than 150 lives*, with hundreds still unaccounted for. The search efforts are ongoing in North Carolina, where the hurricane caused significant destruction and left many people without shelter or basic necessities.

**Qwen2.5-Omni:**      **No answer**

*Well, it's a bit of a tough one. I'm not sure exactly how many people died in the hurricane.* You could try looking at official reports or news articles from the time of the hurricane. Maybe there were some estimates or official counts? *If you want, we could also try to dig deeper into some reliable sources.* What do you think?

Figure 7: MGM-Omni is capable of understanding long-form audio.

**LONG-FORM SPEECH COMPARISON**

🔊 ⦙⫶⦙⦙⫶⦙⫶⦙⫶⦙    Generated Speech Length:    3 min 40 s +

豫章故郡，洪都新府。星分翼轸，地接衡庐。襟三江而带五湖，控蛮荆而引瓯越。物华天宝，龙光射牛斗之墟；人杰地灵，徐孺下陈蕃之榻。雄州雾列，俊采星驰。台隍枕夷夏之交，宾主尽东南之美。都督阎公之雅望，棨戟遥临；宇文新州之懿范，襜帷暂驻。十旬休假，胜友如云；千里逢迎，高朋满座。腾蛟起凤，孟学士之词宗；紫电青霜，王将军之武库。家君作宰，路出名区；童子何知，躬逢胜饯。
……
滕王高阁临江渚，佩玉鸣鸾罢歌舞。画栋朝飞南浦云，珠帘暮卷西山雨。闲云潭影日悠悠，物换星移几度秋。阁中帝子今何在？槛外长江空自流。

**Compare the last 17s**

▲ : Segmentation error    Word : Poor voice cloning    ☐ : Mispronunciation    ~~ : Noise

**OURS**

滕王高阁临江渚，佩玉鸣鸾罢歌舞。画栋朝飞南浦云，珠帘暮卷西山雨。闲云潭影日悠悠，物换星移几度秋。阁中帝子今何在？槛外长江空自流。
▲

**HIGGS**

滕王高阁临江渚，佩玉鸣鸾罢歌舞。画栋朝飞南浦云，▲珠帘暮卷西山雨。闲云潭影日悠悠，物换星移几度秋。阁中帝子今何在？槛外长江空自流。

**MOSS**

~~~~~ ; ~~~~~
~~~~~ ; ~~~~~ ;
~~~~~ ; ~~~~~ ;

**LONG-FORM SPEECH COMPARISON**

🔊 ⦙⫶⦙⦙⫶⦙⫶⦙⫶⦙    Generated Speech Length:    4 min +

Life, thin and light-off time and time again.
生命，一次又一次轻薄过。
……
Prajna Paramita, soon as soon as life be beautiful like summer flowers and death like autumn leaves.
般若波罗蜜，一声一声，生如夏花，死如秋叶。
Also care about what has.
还在乎拥有什么

**Compare the last 18s**

▲ : Segmentation error    Word : Poor voice cloning    ☐ : Mispronunciation    ~~ : Noise

**OURS**

Prajna Paramita, soon as soon as life be beautiful like summer flowers and death like autumn leaves. 般若波罗蜜，一声一声，生如夏花，死如秋叶。Also care about what has. 还在乎拥有什么。

**HIGGS**

~~~~~~~~~~~~
~~~~~~~~~~~~
~~~~~~~~~~~~
~~~~~~~~~~~~
~~~~~~~~~~~~

**MOSS**

~~~~~~~~~~~~
~~~~~~~~~~~~
~~~~~~~~~~~~
~~~~~~~~~~~~
I just say, I wrote another year.

Figure 8: MGM-Omni is capable of correctly generating long-form speech.

