# OpenReview forum: "MGM-Omni: Scaling Omni LLMs to Personalized Long-Horizon Speech"
_ICLR.cc/2026/Conference — ICLR 2026 Conference Withdrawn Submission_

### Official Review · Reviewer_mPeT · 2025-10-22

**Soundness:** 3
**Presentation:** 3
**Contribution:** 2
**Rating:** 2
**Confidence:** 3

**Summary:**

This paper proposes an omni-LLM which can take speech, text and visual modalities as input and can produce text and speech outputs. Audio encoding is based on Qwen2 audio encoder, LLM backbone is initialized from Qwen3 and the speech output is obtained from the output of the trainable TTS adapter to get the speech tokens + audio decoder (to get the mel features) + HiFiGAN vocoder. The paper presents its novelties as 1) a dual track design where the text token outputs are generated first which are then used to generate the chunked speech tokens, 2) chunked decoding to speed up the inference, 3) better long-audio understanding capabilities.
The model training involves a few stages, in the first stage the model is mainly trained for the ASR task and in the next phase audio QA, audio-instruct VQA, and text instruction tuning data are also included. Experiments are performed on various speech tasks demonstrate competitive performance to that of some recent audio LMs or omni LLMs. The largest improvement was observed in the long-form audio tasks where the chunked decoding improves both the quality of the generated speech as well as speeds up the inference.

**Strengths:**

Originality:
- Limited (see the weaknesses section)

Quality:
* Experimental results show competitive performance as compared to other models on many tasks. In particular, the improvements on long-form audio seem large.

Clarity:
* In terms of language, it is written clearly and it is relatively easy to follow the paper.

Significance:
* The paper is relevant to the multimodal LLM community focusing on joint speech and text output.

**Weaknesses:**

Originality:
- Individual components (audio encoder, LLM backbone, TTS adapter, flow matching, HIFIGAN vocoder) have been used in many studies before. One claim of the paper is that the chunked decoding is a novelty. However, it has limited novelty considering other studies which perform interleaved decoding (text tokens followed by a few speech tokens). Also there are other studies that uses a combination of auto-regressive LLM and a non-autoregressive speech decoder, which also propose approaches that resemble the chunked decoding mentioned here.
Some references:
1. Flow-SLM: Joint Learning of Linguistic and Acoustic Information for Spoken Language Modeling (Chou et al., https://arxiv.org/pdf/2508.09350)
2. SpeakStream: Streaming Text-to-Speech with Interleaved Data (Bai et al., https://arxiv.org/pdf/2505.19206)
3. Interleaved Speech-Text Language Models for Simple Streaming Text-to-Speech Synthesis (Yang et al. https://arxiv.org/pdf/2412.16102, Speech + text interleaving with both acoustic and semantic information )

Clarity:
* Experimental evaluation criteria are not well described. There are references to some benchmarks but in some cases, the evaluation method is not clear even after looking at the appendix. For example, in Table 3, a GPT model has been used as a judge but is there a fixed prompt to get those scores between 1-10?
* In the experiments, did the authors tune the pretrained model separately for each task, or is there a single finetuned model that can handle all the speech tasks mentioned in the evaluations for both English and Chinese?

Quality:
* The experimental results show results on some common benchmarks for several tasks such as Librispeech for ASR which look very positive. However, as shown in the WER results of the competing models, the WERs are already so low that, it is hard to interpret if the proposed system performs significantly better. Similarly, how reliable are the LLM as a judge evaluation and what is the sensitivity of the scores so that we can for sure say that the proposed model is significantly better than the other ones? Answering these questions may be out of scope for this paper, however, this may still show lack of rigorous evaluation of the proposed model.

**Questions:**

1) In the experiments, did the authors tune the pretrained model separately for each task, or is there a single finetuned model that can handle all the speech tasks mentioned in the evaluations for both English and Chinese? Please clarify this point.

2) How reliable are the LLM as a judge evaluation and what is the sensitivity of the scores so that we can for sure say that the proposed model is significantly better than the other ones?

3) The long audio evaluation is not well described. What does the success rate mean (as in the caption of Figure 5, "the average success rate across five materials")

---

### Official Review · Reviewer_Fpbi · 2025-10-29

**Soundness:** 3
**Presentation:** 3
**Contribution:** 2
**Rating:** 4
**Confidence:** 4

**Summary:**

The paper presents MGM-Omni, a model for omni-modal understanding and long speech generation, based on a "brain-mouth" architecture. The "brain" is Qwen2.5-VL, utilizing two audio encoders (Qwen2-Audio and Belle-Whisper), while the "mouth" is Qwen3 with a TTS-Adapter appended to its output. The "mouth" (SpeechLM) converts text into CosyVoice2 speech tokens, followed by flow-matching and HiFi-GAN to generate audio waveforms. The authors introduce a chunk-based parallel decoding scheme to address the text-speech token-rate gap, enabling faster inference and supporting long speech generation (10 minutes). Experiments demonstrate competitive results in ASR, audio QA, and short TTS, with improved performance in long-form audio understanding, VQA and long TTS tasks.

**Strengths:**

1. The chunk-based parallel decoding scheme effectively narrows the text-speech token-rate gap, providing a useful technique for the community.
2. MGM-Omni outperforms baselines in long-form audio understanding and TTS tasks.
3. The focus on omni-modal understanding and speech generation addresses a key research challenge.

**Weaknesses:**

1. The paper does not evaluate on common Spoken QA benchmarks (e.g., OpenAudioBench, VoiceBench, UltraEval-Audio, and Big Bench Audio), leaving the model's Spoken QA performance unclear.
2. Many standard Audio Understanding benchmarks (e.g., MMAU, MMSU, MMAU-Pro) are omitted, making the model's overall Audio Understanding capabilities uncertain.
3. The training pipeline is confusing. Section 3.1 discusses two-phase audio understanding, while Section 3.3 covers two-phase Omni Voice training. A clearer explanation of the full training process, data types, and data sizes is needed. If open-source datasets are used, these should be explicitly stated for reproducibility.
4. Long-form TTS, one of the paper’s main contributions, lacks subjective MOS evaluations.
5. Chunk-based parallel decoding, one of the core innovations, does not clearly explain how chunks are divided.

**Questions:**

Please address the issues described in the Weaknesses section. Resolving these concerns could improve the paper’s evaluation.

---

### Official Review · Reviewer_sa1i · 2025-10-29

**Soundness:** 2
**Presentation:** 2
**Contribution:** 2
**Rating:** 4
**Confidence:** 4

**Summary:**

This paper presents MGM-Omni, an open-source omni-modal large language model (LLM) designed for unified understanding and generation across text, image, video, and audio modalities. The model features a dual-track "brain–mouth" architecture, separating multimodal reasoning (MLLM) from speech synthesis (SpeechLM), and introduces chunk-based parallel decoding for efficient long-form speech generation. MGM-Omni demonstrates strong performance on a range of benchmarks, including long-form audio understanding, zero-shot voice cloning, and long-horizon speech generation, and introduces a new Long-TTS-Eval benchmark for systematic evaluation.

**Strengths:**

1. The paper presents a well-engineered system that integrates state-of-the-art components for both audio understanding and speech generation.
2. The introduction of the Long-TTS-Eval benchmark is a useful addition for the community to evaluate long-form TTS systems.
3. The commitment to open-sourcing the model and benchmarks is valuable for reproducibility and further research.

**Weaknesses:**

1. The core architectural idea—the dual-track "brain–mouth" design—is essentially the same as the "thinker–talker" architecture already established in Qwen-Omni and related works. The paper does not sufficiently differentiate its approach or provide substantial conceptual innovation beyond prior art.
2. The contributions are primarily engineering improvements and system integration, rather than introducing fundamentally new algorithms or modeling paradigms.
3. The system is effectively a combination of two models (ASR for understanding, TTS for generation) rather than a fully end-to-end multimodal model. The separation between MLLM and SpeechLM, while practical, does not advance the field toward unified end-to-end learning.
4. While the empirical results are strong, the improvements over existing models (especially Qwen2.5-Omni and CosyVoice2) are incremental and largely attributable to system scaling and data engineering, rather than novel techniques.

**Questions:**

1. Your system is described as a unified omni-modal LLM, but in practice it seems to be a combination of an ASR module for understanding and a TTS module for generation. How does your approach move beyond simply integrating ASR and TTS, and what steps have you taken toward true end-to-end learning?
2. Can you provide more detailed ablation studies to isolate the impact of each proposed component (e.g., chunk-based decoding, dual audio encoders) and demonstrate their necessity?

---

### Official Review · Reviewer_giy4 · 2025-10-31

**Soundness:** 3
**Presentation:** 2
**Contribution:** 2
**Rating:** 4
**Confidence:** 5

**Summary:**

This work aims to address the insufficient modeling of long-form audio understanding and long-form speech generation with consistent timbre and prosody, among existing multimodal large language models. The proposed MGM-Omni model incorporates a dual-track architecture to separate the MLLM-based multimodal understanding and reasoning from SpeechLM for real-time speech generation. Specifically, to integrate audio understanding capabilities into a VLM, MGM-Omni uses dual audio encoder and multi-stage training of audio-to-text pre-training and omni-modal training; and uses chunk-based parallel decoding to tackle the token-rate discrepancy between speech and text and improve multimodal alignment and long-form speech generation. The paper also introduces a long-form TTS benchmark Long-TTS-Eval. Experimental results demonstrate strong long audio understanding and long-form TTS capabilities of MGM-Omni.

**Strengths:**

1.	The two key innovations are reasonable and could be explored in other MLLMs for improving long-form generation.

a.	The chunk-based decoding design segments input text to SpeechLM into smaller chunks, and each chunk produces a corresponding speech segment. The common speech token delay strategy is applied within each chunk. This chunk-based decoding strategy reduces the distances between speech and text modalities and mitigates early mis-synchronization, hence effectively improving speech-text alignment for long-form generation, since for long-form generation, the speech-text mis-synchronization could be more severe.

b.	The parallel decoding with FSQ speech tokenizers further improves speech-text alignment and  also improves efficiency.

2.	The proposed MGM-omni is evaluated on ASR, audio understanding, omni-understanding, and short and long TTS.

**Weaknesses:**

1.	Missing related works: Some important related works are not cited, discussed or compared.

a.	For long-form speech generation, VibeVoice (open-sourced in August 2025) can generate up to 90 mins speech with up to 4 distinct speakers, but it is not discussed or compared in the paper.

b.	On ASR and audio understanding capabilities, some strong LALMs and omni models are missing. The LALM Kimi-audio (open-sourced in April 2025) and the omni model Ming-Lite-Omni (open-sourced in May 2025) are not compared theoretically and empirically. For example, the WER of kimi-audio (7B) and the Ming-Lite-omni on Librispeech clean/other are 1.28/2.42 and 1.44/2.8, as reported in their tech report, which are all lower than MGM-Omni-32B’s 1.5/3.2. For audio understanding capabilities, kimi-audio and Ming-Lite-omni reported impressive performance on a series of audio understanding benchmarks.  Some commonly used audio understanding benchmarks include OpenAudioBench, VoiceBench, Big Bench Audio etc. This work uses AIR-Bench, a generative comprehension benchmark for LALMs with LLM-as-judge. It would be useful to conduct more comprehensive audio understanding evaluations in addition to the used AIR-Bench, and include strong LALMs such as kimi-audio, MiniCPM-o 2.6 and omni models such as Ming-Lite-omni into comparison.

2.	The motivations of some algorithmic designs need to be clearly explained:

a.	Please clarify the rationale of the parallel decoding design, how it compares with separate text and speech heads in multi-token prediction. The ablation study shows that with parallel 4, RTF is notably improved with a tradeoff on WER.

b.	The benefit of incorporating the additional Whisper-based encoder Belle-Whisper-large-v3 with the main Whisper-large-v3 based qwen2-audio encoder needs to be clarified. The paper mentioned that the additional encoder is specialized in Chinese ASR. It is interesting to see that in the ablation study of the audio encoders, as shown in Table 6a on CommonVoice ASR, integrating the two audio encoders based on Info Mining improved Chinese CER from 3.9 to 3.5 (10%), yet a more substantial improvement on English WER (30% relative). Hence, the strengths of the additional Belle-Whisper-large-v3 need to be further clarified for the contribution.

c.	 How does the Omni length understanding training strategy compare to other curriculums, such as training on same, shorter length samples first and then training on variable lengths with same-length batches and dynamic batch sizes as used in the paper?

3.	Some implementation details are missing, posing difficulity to reproducibility. For example, details of the pre-training data stats of the two stages for audio understanding are missing, and details of  the post-training data stats of SpeechLM is not specified, details of hyperparameters are not provided.

4.	Some empirical validations are insufficient:

a.	For long-form audio understanding, the evaluation is based on a needle-in-the-haystack test over only 5 videos, comparing to qwen2.5-omni. This is not a rigorous and systematic evaluation. Long audio samples could be extracted from existing audio benchmarks, such as the DESED dataset or the RACE dataset (large-scale reading comprehension dataset) synthesized by TTS.

b.	As suggested earlier, it would be useful to incorporate the commonly used audio understanding benchmarks into evaluation.

c.	As suggested earlier, it is important to include missing strong open-sourced LALMs such as Kimi-audio and omni models such as Ming-lite-omni into audio understanding, omni understanding, and VibeVoice into speech generation comparisons.

d.	Since audio understanding is incorporated through multi-training stages, it is important to compare the VL understanding capabilities between MGM-Omni and the original VLM, to assess forgetting.

e.	It is important to add statistical significance tests on the gains from MGM-Omni over the best baselines, since some of them seem to be small.

**Questions:**

Please address the questions listed under Weaknesses.

---

### Note · Authors · 2025-11-14

I have read and agree with the venue's withdrawal policy on behalf of myself and my co-authors.